# Differential activation of a frontoparietal network explains population-level differences in statistical learning from speech

Joan Orpella[1], M. Florencia Assaneo[2]*, Pablo Ripollés[1,3,4,5], Laura Noejovich[1], Diana López-Barroso[6,7], Ruth de Diego-Balaguer[8,9,10,11]‡, David Poeppel[1,4,5,12]‡

**1** Department of Psychology, New York University, New York, New York, United States of America, **2** Institute of Neurobiology, National Autonomous University of Mexico, Juriquilla, Querétaro, Mexico, **3** Music and Audio Research Lab (MARL), New York University, New York, New York, United States of America, **4** Center for Language, Music and Emotion (CLaME), New York University, New York, New York, United States of America, **5** Max Planck Institute for Empirical Aesthetics, Frankfurt, Germany, **6** Cognitive Neurology and Aphasia Unit, Centro de Investigaciones Médico-Sanitarias, Instituto de Investigación Biomédica de Málaga–IBIMA and University of Málaga, Málaga, Spain, **7** Department of Psychobiology and Methodology of Behavioral Sciences, Faculty of Psychology and Speech Therapy, University of Málaga, Málaga, Spain, **8** ICREA, Barcelona, Spain, **9** Cognition and Brain Plasticity Unit, IDIBELL, L'Hospitalet de Llobregat, Barcelona, Spain, **10** Department of Cognition, Development and Educational Psychology, University of Barcelona, Barcelona, Spain, **11** Institute of Neuroscience, University of Barcelona, Barcelona, Spain, **12** Ernst Struengmann Institute for Neuroscience, Frankfurt, Germany

☯ These authors contributed equally to this work.
‡ These authors jointly supervised this work.
* fassaneo@gmail.com

**Data Availability Statement:** All relevant data can be found in the paper's Supporting Information files.

## Abstract

People of all ages display the ability to detect and learn from patterns in seemingly random stimuli. Referred to as statistical learning (SL), this process is particularly critical when learning a spoken language, helping in the identification of discrete words within a spoken phrase. Here, by considering individual differences in speech auditory–motor synchronization, we demonstrate that recruitment of a specific neural network supports behavioral differences in SL from speech. While independent component analysis (ICA) of fMRI data revealed that a network of auditory and superior pre/motor regions is universally activated in the process of learning, a frontoparietal network is additionally and selectively engaged by only some individuals (high auditory–motor synchronizers). Importantly, activation of this frontoparietal network is related to a boost in learning performance, and interference with this network via articulatory suppression (AS; i.e., producing irrelevant speech during learning) normalizes performance across the entire sample. Our work provides novel insights on SL from speech and reconciles previous contrasting findings. These findings also highlight a more general need to factor in fundamental individual differences for a precise characterization of cognitive phenomena.

## Introduction

Statistical learning (SL) is the capacity to use distributional information present in the environment to extract meaningful regularities. SL has been demonstrated across age groups from birth [1,2], sensory modalities (e.g., audition [3,4], vision [5], and touch [6]), representational

**Funding:** This work was supported by National Institutes of Health (https://www.nih.gov/) grant 2R01DC05660 (D.P.), National Science Foundation (https://www.nsf.gov/) grant 2043717 (P.R., D.P., J.O. and M.F.A.), European Research Council (https://erc.europa.eu/) grant ERC-StG-313841 (TuningLang; RdD-B), and Spanish Ministry of Science and Innovation (https://www.ciencia.gob.es/) grant BFU2017-87109-P (RdD-B), which is part of Agencia Estatal de Investigación (AEI) (Co-funded by the European Regional Development Fund. ERDF, a way to build Europe). We also thank the CERCA Program / Generalitat de Catalunya for the institutional support. DL-B was supported by a travel grant from the University of Málaga and the Ramón y Cajal program (RYC2020-029495-I) funded by the Spanish Ministry of Science and Innovation (MCIN/AEI/10.13039/50110001103) and FSE (invierte en tu futuro). The funders had no role in study design, data collection and analysis, decision to publish, or preparation of the manuscript.

**Competing interests:** The authors have declared that no competing interests exist.

**Abbreviations:** AIC, Akaike information criterion; AS, articulatory suppression; FDR, false discovery rate; FWE, family-wise error; GIFT, Group ICA of fMRI Toolbox; GLM, generalized linear model; ICA, independent component analysis; IFG, inferior frontal gyrus; IPhOD, Phonotactic Online Dictionary; MNI, Montreal Neurological Institute; PL, passive listening; PLV, phase locking value; SL, statistical learning; SSS test, Spontaneous Speech Synchronization test; STG, superior temporal gyrus; SWFL, statistical word form learning.

domains [5] (temporal and spatial), and even species [7,8]. In the domain of speech and language processing, statistical word form learning (SWFL) is considered critical in the early stages of language acquisition as the ability to segment phonological word forms from continuous speech [3,9]. Segmented word forms are readily associated with meanings [10] and can also be used in subsequent stages to discover grammatical relationships [11]. While regarded fundamental to language learning by most contemporary theories, the precise neural substrates of this ubiquitous phenomenon are not well understood and remain controversial.

There have been several experimental attempts to pinpoint the neural basis of SWFL, but the existing literature shows inconsistent results. Some studies report a correlation between SWFL performance and the activity of the superior temporal gyrus (STG) and dorsal pre/motor regions [12–15]. Other experiments instead implicate the left inferior frontal gyrus (IFG) [16] and its interaction with superior temporal areas [15,17]. We hypothesize that the inconsistency of results in the literature is a consequence of very specific individual differences in the neural resources allocated for SWFL.

In a recent study, we provided a first glance of how one might capitalize on individual differences to gain deeper mechanistic insights into SWFL: Individual listeners grouped by their spontaneous speech auditory–motor synchronization abilities turn out to differ in their SWFL performance [18]. Specifically, we showed there a behavioral task (the Spontaneous Speech Synchronization test, henceforth "SSS test") that robustly classifies participants into high and low speech auditory–motor synchronizers. At the brain level, high synchronizers showed a greater brain-to-stimulus synchrony in the left IFG during passive speech listening as well as more volume in the white matter pathways underlying the dorsal language stream (i.e., the arcuate fasciculus) [19]. Critically, the high/low synchronizer distinction was predictive of SWFL performance (Fig 1A), but the relationship between auditory–motor synchrony, population-level brain differences, and SWFL remained elusive. Here, we hypothesized that the dorsal language stream, including the IFG, is not only linked to auditory–motor synchrony as previously reported but also gives high synchronizers the advantage in SWFL over low synchronizers.

To test this hypothesis, we used independent component analysis (ICA) of fMRI data in combination with a classic behavioral paradigm designed to interfere with the availability of the dorsal language stream for SWFL [20]. Specifically, the behavioral paradigm employed involves the contrast between passive listening (PL) and articulatory suppression (AS) conditions (Fig 1B). AS requires participants to repeat a nonsense syllable during (word) learning, which hampers SWFL performance [9]. ICA, on the other hand, is a data-driven neuroimaging approach well suited to identify spatially independent and temporally coherent brain networks that support specific cognitive processes [21]. Previous work using this approach has related SWFL to a network comprising auditory and superior pre/motor areas [14]. This earlier work, however, did not consider the high/low synchronizer distinction. In all, in the current experiment we investigate, in both high and low synchronizers, the brain networks engaged during SWFL under PL and AS conditions as well as the behavioral consequences of AS for performance (Fig 1). We hypothesize that, if high synchronizers show better learning—putatively due to a greater reliance on the dorsal language stream—they should show a greater recruitment of this functional anatomic stream than low synchronizers during learning as well as a greater AS effect.

## Results

### Behavioral results: AS modulates high but not low synchronizers' SWFL performance

An initial cohort (*N* = 55, 34 females; mean age, 22; age range, 18 to 37) underwent behavioral testing. Participants completed 4 blocks of statistical word learning in 2 different experimental

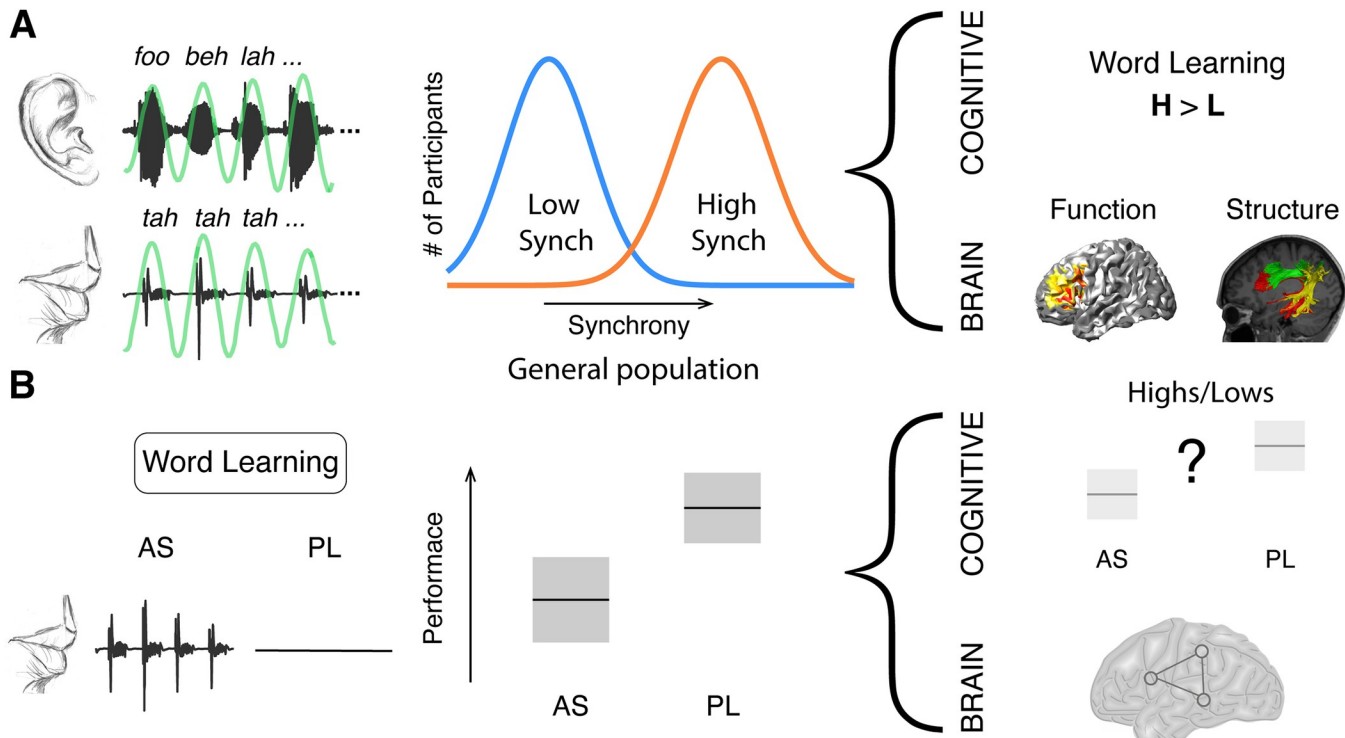

**Fig 1. Previous work motivating the hypothesis. (A)** During the SSS test, participants listen to an isochronous stream of random syllables (rate 4.5 syllables/ sec) while concurrently whispering the syllable "tah." Left panel: example of the perceived (upper panel) and produced (lower panel) signals. Green line, band- pass filtered envelope used to compute input-output synchrony. Middle panel: Synchrony between perceived and produced syllables yields a bimodal distribution, allowing the classification of participants into low (blue) and high (orange) synchronizers. While some participants spontaneously align the produced syllabic rate to the perceived one (high synchronizers), others show no modification of the produced rate due to the presence of the external rhythm (low synchronizers). Right panel: High synchronizers outperformed lows in a statistical word learning task. They also showed enhanced brain-to-speech synchronization over left frontal regions and a greater volume in the white-matter pathways connecting temporal and frontal areas [18]. **(B)** The word learning task consists of a learning phase wherein 4 trisyllabic pseudo-words are presented in a continuous stream. Learning is assessed post exposure. Left panel: Participants are instructed to repeat a nonsense syllable (AS condition) or to passively listen (PL condition) during the entire learning phase. Middle panel: predicted performance decreases due to AS [9]. Right panel: differences between high and low synchronizers are hypothesized at the cognitive and brain levels. AS, articulatory suppression; PL, passive listening.

conditions, PL and AS, followed by the SSS test (Methods and Fig 1). In both tasks, the audi- tory stimuli were presented at a rate of 4.5 syllables per second, corresponding to the mean syl- lable rate across languages [22–24] and the natural frequency of speech motor regions [25]. The outcome of the SSS test showed the expected [18,26] bimodal distribution, allowing the classification of participants into high and low synchronizers (Fig 2A). Moreover, the syn- chrony between perceived and produced syllables in the SSS test was highly correlated with that in the AS blocks (Fig 2B; $N = 55$, Spearman correlation coefficient r = 0.75, $p < 0.001$). This demonstrates that speech-to-speech synchrony is not only reliable across time, as was pre- viously demonstrated [18], but also across tasks, confirming that auditory–motor synchrony is a stable feature of each individual.

A linear mixed-model analysis ($N = 55$; see Methods) of the learning performance showed a significant decrement in AS relative to PL (Fig 2C; Main effect of Condition: $\chi2 = 15.4$, $p < 0.001$). This result thus replicates previously reported AS effects on SWFL [9]. The analysis also showed a main effect of Group (Highs > Lows; $\chi2 = 9.11$, $p < 0.01$) in line with our previ- ous work [18] and, importantly, a significant interaction between the 2 factors

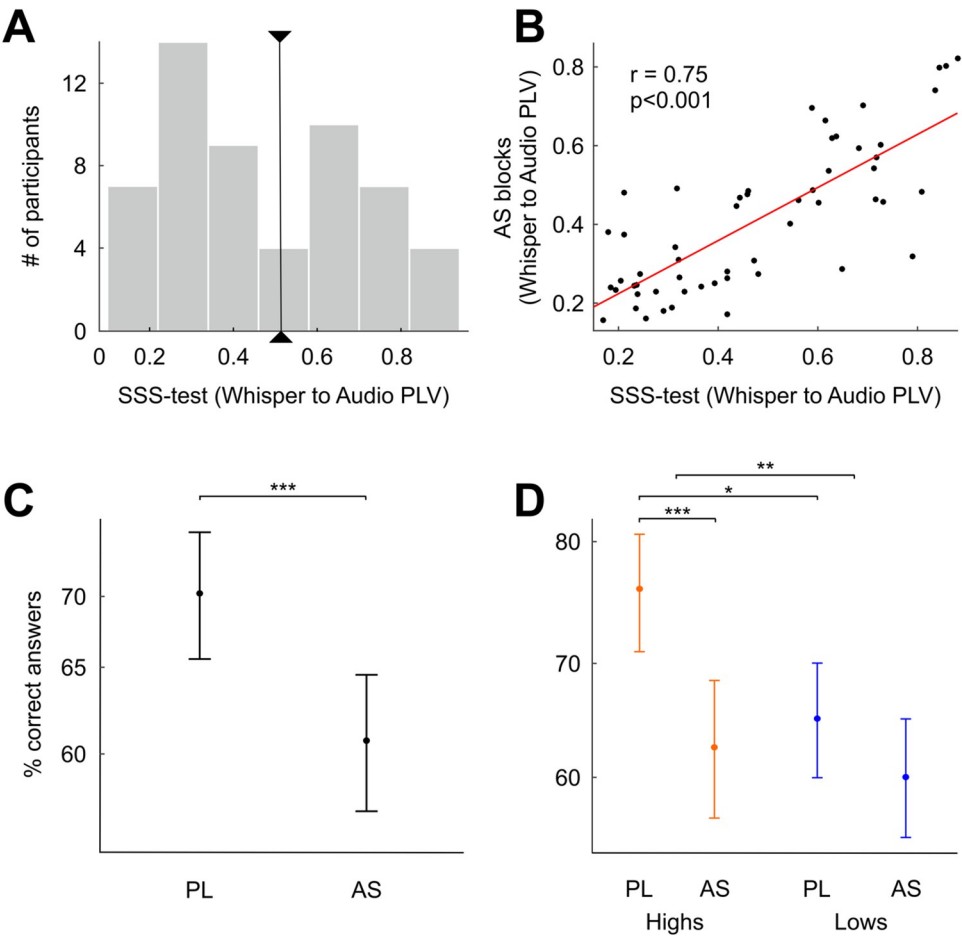

**Fig 2. AS modulates only high synchronizers' performance. (A)** SSS test outcome. Histogram of the PLVs (measure of speech-to-speech synchrony; see Methods) between the envelope of the perceived and produced speech signals, band-pass filtered at 3.5 to 5.5 Hz. Black line, critical value separating high and low synchronizers ($N = 55$; see Methods). **(B)** Participants' PLV during AS as a function of the PLV from the SSS test. Red line represents the correlation of the data. **(C)** Percentage of correct responses during PL and AS across the entire sample. **(D)** Percentage of correct responses during PL and AS for the low (blue) and the high (orange) synchronizers. $^{*}p < 0.05$, $^{**}p < 0.01$, $^{***}p < 0.001$ Linear mixed model results. Dots: model predicted group means. Bars: 95% confidence interval. Data for Fig 2A and 2B can be found in S1 Data. Data for Fig 2C and 2D can be found in S2 Data. AS, articulatory suppression; PL, passive listening; PLV, phase locking value; SSS test, Spontaneous Speech Synchronization test.

(Condition*Group; $\chi 2 = 4.22$, $p < 0.05$). Critically, when the sample was next split into high and low synchronizers by estimating their corresponding marginal means (see Methods), we observed the AS effect in the population of high synchronizers (Fig 2D; $N_{high} = 23$, zratio = 3.92, $p < 0.001$) but not in the population of low synchronizers (Fig 2D; $N_{low} = 32$, zratio = 1.63, $p = 0.1$); that is to say, the performance of low synchronizers was not modulated by the action of speaking during the learning phase. Additionally, in line with previously reported data [18], high synchronizers outperformed lows in the PL condition (zratio = 3.02, $p < 0.01$), but there was no difference in performance between groups in the AS condition (zratio = 0.64, $p = 0.52$). Importantly, for all groups and conditions, learning remained significantly above chance (signed rank tests against chance level, 2-sided: $p_{high/AS} < 0.001$, $p_{high/PL} < 0.001$, $p_{low/AS} < 0.001$, $p_{low/PL} < 0.001$).

## Neuroimaging results (I): High synchronizers activate an additional brain network during statistical word learning

Having established the expected behavioral differences between high and low synchronizers, we next acquired fMRI data from a new group of participants ($N = 41$) while they performed the same behavioral paradigm (see Methods). The SWFL paradigm was optimized for fMRI testing. Specifically, we included both a rest block and a speech motor block as control conditions. During the speech motor block, participants were required to repeatedly whisper the syllable "tah" with no concurrent auditory input. The behavioral performance in the scanner showed the same trend as the learning pattern obtained with the first sample (S1 Fig; Condition (PL > AS): $\chi 2 = 5.40$, $p < 0.05$; Group (Highs > Lows): $\chi 2 = 3.67$, $p = 0.055$; Condition*Group: $\chi 2 = 2.74$, $p = 0.098$; Highs (PL > AS): zratio = 2.32, $p < 0.05$; Lows (PL versus AS): zratio = 0.08, $p = 0.93$; Highs versus Lows in PL: zratio = 1.92, $p = 0.055$; Highs versus Lows in AS: zratio = 0.05, $p = 0.96$). Even under notably adverse listening/learning conditions (i.e., during fMRI scanning), the detrimental effect of the AS condition was restricted to the high synchronizers.

Using the Group ICA of fMRI Toolbox (GIFT; see Methods) [21], we identified 5 brain networks that were significantly recruited during SWFL in the PL and/or the AS condition (S2 and S3 Figs). Critically, a frontoparietal network including bilateral inferior and middle frontal gyri, inferior parietal cortex, and the supplementary motor area distinguished between high and low synchronizers during the PL condition (Fig 2A; $N_{high} = 18$, $N_{low} = 20$, Mann–Whitney–Wilcoxon test, 2-sided $p = 0.038$, false discovery rate [FDR] corrected). Moreover, while the activity of this network during PL was statistically significant for high synchronizers, it was not for the lows (Mann–Whitney–Wilcoxon test, 2-sided $p_{high} < 0.005$ and $p_{low} = 0.9$, respectively, FDR corrected). Moreover, we found moderate evidence in favor of the null hypothesis that the network was not activated during PL for the lows (Bayes Factor $BF_{01} = 4$).

Similarly, during AS, only high synchronizers significantly engaged the frontoparietal network (Fig 3A; Mann–Whitney–Wilcoxon test, 2-sided $p_{high} < 0.005$ and $p_{low} = 0.42$, FDR corrected), again with moderate evidence in favor of the null hypothesis for the lows ($BF_{01} = 4.1$). In this condition, however, the network's activity did not differentiate between the groups. Given that groups were defined by their speech auditory–motor synchrony, we then correlated the engagement of the frontoparietal network with the synchronization (phase locking value, PLV) between the perceived and produced syllables during AS. Indeed, these measures were positively correlated in the entire sample as well as in the high synchronizers only (Fig 3B; Spearman correlation coefficient $r_{all} = 0.41$ and $r_{high} = 0.56$, $p_{all} = 0.009$ and $p_{high} = 0.012$), suggesting a link between spontaneous auditory–motor synchrony and frontoparietal network engagement.

## Neuroimaging results (II): The interplay between networks boosts learning

Next, we assessed whether the activity of any of the networks significantly engaged during the PL condition was predictive of SWFL. Replicating previous results [14,27], we found a network comprising mainly bilateral auditory regions and a small superior pre/motor cluster (henceforth, auditory network) whose activity positively correlated with learning performance in the whole sample (Fig 4A; Spearman correlation coefficient r = 0.42 and $p = 0.032$, FDR corrected). We found no significant correlations between learning and network activity in the AS condition. Since during PL (i) highs behaviorally outperformed lows; (ii) the frontoparietal network was only activated by high synchronizers; and (iii) the auditory network was related to learning performance, we next examined whether the learning benefit of high synchronizers over lows in SWFL was related to the interaction between the 2 networks (auditory and

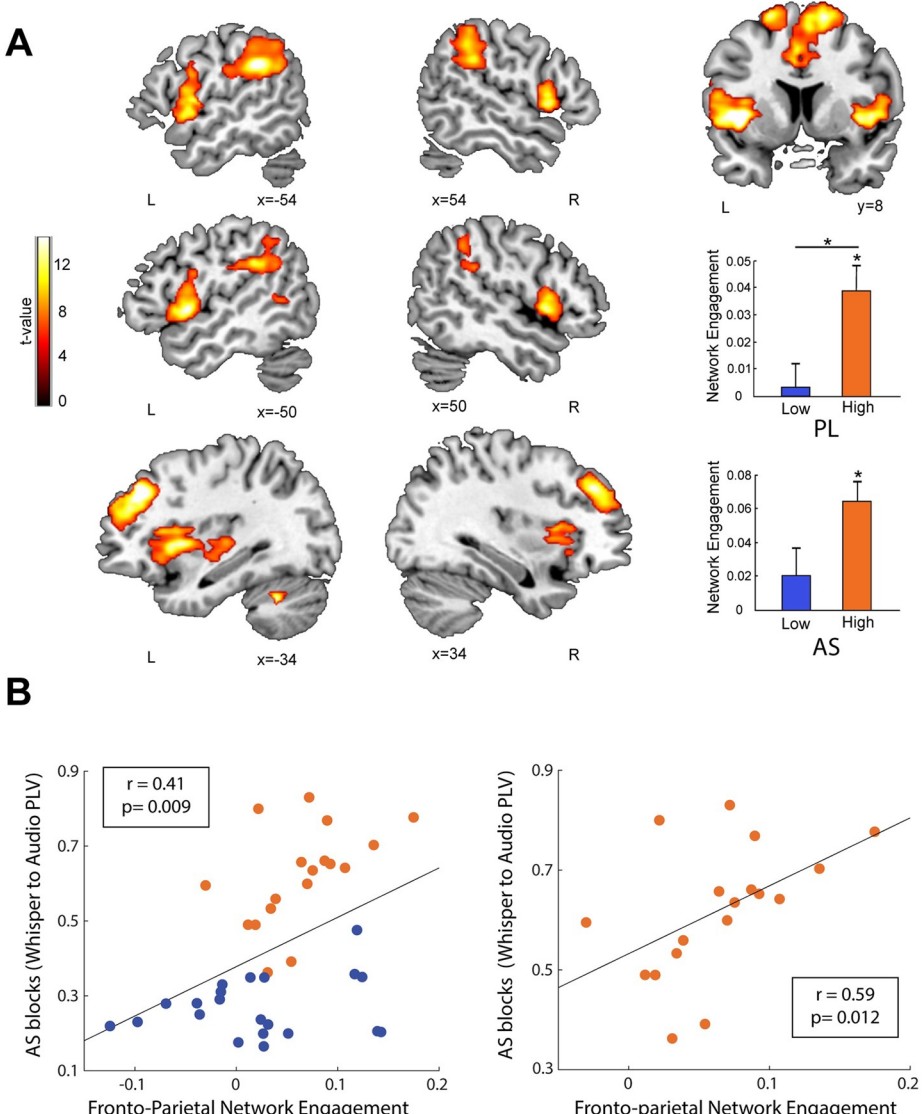

**Fig 3. High synchronizers activate an additional brain network during statistical word learning. (A)** In red/yellow, the frontoparietal network is shown over a canonical template, with MNI coordinates on the lower portion of each slice. Neurological convention is used. A $p < 0.05$ FWE-corrected threshold at the cluster level with an auxiliary $p < 0.001$ threshold at the voxel is used. This is the only network showing significant differences in activity between high and low synchronizers during PL (see bar plots on the lower right; $^*$ $p < 0.05$, FDR corrected). **(B)** Scatterplot displaying participants' PLV during AS as a function of the frontoparietal network's engagement. Black line represents the correlation of the data. Left panel: all participants. Right panel: high synchronizers. Data for Fig 3A (top bar plot) can be found in S7 Data. Data for Fig 3A (bottom bar plot) and for Fig 3B can be found in S3 Data. FDR, false discovery rate; FWE, family-wise error; MNI, Montreal Neurological Institute; PL, passive listening; PLV, phase locking value.

frontoparietal). Specifically, we explored the relationship between the time courses of these 2 networks at the individual listener level and the learning benefit. As illustrated in Fig 4B, high synchronizers with a greater learning benefit (defined as PL minus AS) appeared to show a distinct pattern with interweaving time courses between the auditory and frontoparietal networks. To quantify this observation, we employed an analysis typically used in electronics: XOR. Applied to our signals, this logical operation assigns a single value per time point: one

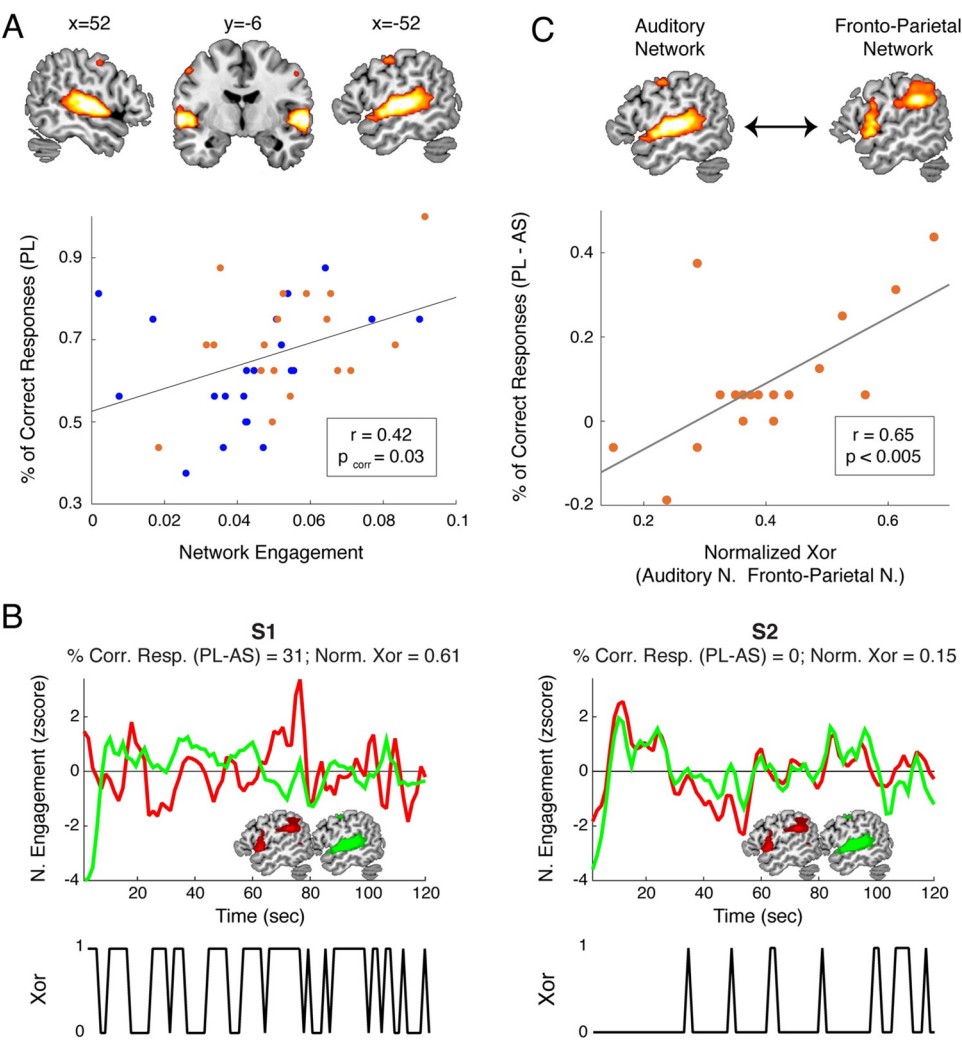

**Fig 4. An interplay between networks boosts learning. (A)** The auditory network supports learning during PL. Upper panel: In red/yellow, the auditory network is shown over a canonical template, with MNI coordinates on the upper portion of each slice. Neurological convention is used with a $p < 0.05$ FWE-corrected threshold at the cluster level, with an auxiliary $p < 0.001$ threshold at the voxel level. Lower panel: Scatterplot displaying participants' percentage of correct responses during PL as a function of the auditory network's engagement. **(B)** The learning benefit during PL (correct answers in PLcorrect answers in AS) is related to the interplay between the time courses of the frontoparietal (red) and the auditory (green) networks. Left/right panel: a representative high synchronizer with a greater/smaller learning benefit. Lower panels: Time evolution of the XOR analysis. **(C)** Scatterplot displaying high synchronizers' learning benefit as a function of the normalized XOR between the frontoparietal (red) and the auditory (green) networks. Red line: correlation of the data. Data for Fig 4B and 4C can be found in S4 Data. Data for Fig 4A can be found in S7 Data. FWE, family-wise error; MNI, Montreal Neurological Institute; PL, passive listening.

[1] when a single network is above baseline activity, or zero (0) otherwise; that is, a one is assigned when one or the other network (but not both) is active (Fig 4B, lower insets). For each high synchronizer, we averaged the XOR over time, and correlated this value with their learning benefit (PL-AS) (note that this analysis would be meaningless for low synchronizers, given the nonsignificant activation of their frontoparietal network). In line with the observed pattern, a positive correlation was found (Fig 4C; Spearman correlation coefficient r = 0.65, $p < 0.005$). This suggests that the learning benefit shown by high synchronizers over lows is

related to a specific pattern of activity highlighted by the XOR rather than a perfect correlation between time courses of the networks.

## Discussion

The behavioral and neuroimaging data show that the neural substrates supporting SWFL vary across individuals in a systematic way. We arrived at this observation by splitting the population into 2 groups according to their spontaneous speech auditory–motor synchronization abilities (Fig 2), a classification that has now been shown to be robust in a number of experiments, both in-lab and online, in different languages, and with different experimental manipulations [18,26]. Specifically, we found 2 distinct networks related to SWFL performance. One network encompasses mainly auditory regions and a small superior pre/motor cluster (auditory network), appears to be universally or generically recruited, and directly correlates with learning. Another network, including inferior frontal, inferior parietal and supplementary motor areas (frontoparietal network), is neither necessary nor sufficient for learning, yet it boosts learning performance. This latter network, whose activity correlates with spontaneous auditory–motor synchrony, is exclusively recruited by high auditory–motor synchronizers during learning. These observations parsimoniously account for the apparently disparate results in previous SWFL literature and provide a new way to discuss SL in neural terms.

In terms of behavior, we find that the effects of AS are not universal. Typically, the execution of an AS task leads to performance deficits. We demonstrate—in 2 independent cohorts—that only the performance of participants with a high degree of auditory–motor synchronization is affected by AS. Low synchronizers, in contrast, remain unaltered in their word learning performance. Our results thus indicate that articulatory rehearsal is not necessary for SWFL but its additional recruitment confers a learning benefit: high synchronizers, who show robust AS effects, performed better than lows during PL. Note that these results are not discordant with the previous literature [9] since averaging across high and low synchronizers yields the expected overall AS effects.

At the neural level, we found an important distinction between high and low synchronizers with respect to the engagement of a frontoparietal network: Only high synchronizers engage this network during PL. While SWFL performance correlates with the activity of the auditory network across the entire sample—in line with previous literature [14]—a synergistic relationship between both networks boosts learning performance in the high synchronizer group. Importantly, the engagement of the frontoparietal network also predicted the degree of spontaneous synchronization of produced speech during the AS condition.

A relationship between auditory–motor synchronization and language skills has been previously reported in the literature [18,28,29]. For example, precision in tapping to a metronome has been argued to correlate with reading and spelling abilities in children with developmental dyslexia [30]. Similarly, synchronization to a beat correlates with phonological awareness and rapid naming in typically developing preschoolers [31]. Despite the cumulative evidence for a link between auditory–motor synchronization and these various language skills, the existence of a neural substrate shared among these apparently unrelated cognitive abilities remains an empirical question. With this question in mind, our results suggest that the reported fronto-parietal cortical network subserves this shared role: On the one hand, the engagement of this network during PL confers a benefit in learning; on the other, during AS, the engagement of this network predicts the degree of speech auditory–motor synchronization.

Insofar as there are differences between high and low synchronizers at a structural level and differences in synchrony that are stable in time [18], we understand the high/low synchronizer differences reported in this and previous works as trait differences. We have also theorized on

how structural connectivity differences can give rise to the synchrony differences between the groups under particular stimulation conditions (e.g., auditory stimulation within a specific frequency range [32]). Because of this, our reported group differences could also be understood as state differences, the precise functional significance of which remains an open question.

From a mechanistic and more neurophysiologically motivated perspective, we propose that enhanced syllable-level segmentation or parsing—a key prerequisite for SWFL—results from the coordinated activity between auditory and frontoparietal networks, ultimately leading to better SWFL performance. In line with this conjecture, we previously showed that the frontal—most component of the frontoparietal network, the IFG, aligns with the onset of passively perceived syllables in high synchronizers. This frontal region has also been shown to send top-down signals to auditory cortex (the main component of the auditory network) to better align its activity to the speech input [33,34]. A similar proposal has been advanced in the literature to account for the enhanced processing of phonology that results from improved auditory timing perception through auditory–motor training [35]. Broadly speaking, therefore, our results line up with recent theories of SL, which postulate the works of both learning systems (e.g., comprising auditory areas in the case of auditory input) and modulatory attentional/control systems (e.g., as supported by frontoparietal networks) underlying learning performance [36,37]. However, we add to these current views of SL by specifying the role of these modulatory systems in terms of timing operations critical to auditory–motor synchronization advantageous to learning.

On the other hand, there exist multiple and distinct frontoparietal networks associated with attention that overlap with our reported frontoparietal network in high synchronizers. Its ventral frontoparietal component (i.e., inferior prefrontal to inferior parietal cortex), for example, has been related to stimulus-driven attention [38], which may in turn be related to the salience network [39]. Note that the stimulus-driven attention network is mostly bilateral (e.g., [40]) but shows different patterns of lateralization, rightward for spatial attention and leftward for temporal attention [41]. Given the relationship between our frontoparietal network and auditory–motor synchronization (this paper and [18]), a possibility therefore is that high synchronizers' frontoparietal engagement relates to a temporal attention mechanism.

Another possibility is that frontoparietal activity in high synchronizers relates to a control network (e.g., [42]) that flexibly interacts with other task-specific networks (e.g., [43,44]). This is possible given the activity in more dorsal frontal regions that also feature in our frontoparietal network. Interestingly, recent articles (e.g., [45,46]) show that a supra-modal frontoparietal network entrains to stimulation (sensory and via rhythmic transcranial magnetic stimulation) in the theta band and that this entrainment (causally) enhances auditory working memory. This is very similar to our previous [18] and current findings, in which high synchronizers entrain to theta stimulation (higher behavioral PLV and brain-stimulus PLV during PL) and show a behavioral advantage over individuals that do not show this entrainment. The extent to which this and the aforementioned frontoparietal networks are one same network or different networks that interact for high synchronizers during the SL task cannot be answered by our current data and so remains an empirical question. However, our analysis (ICA for fMRI) indicates that, at the very least, these frontoparietal regions' time-courses cohere in time.

There are also reasons to distinguish these frontoparietal networks from the dorsal network for goal-directed attention [47], despite a similar involvement of dorsal prefrontal regions. In contrast to research showing SL benefits from interfering with this network (e.g., [48,49]), we show that AS hinders learning. Moreover, frontoparietal involvement, which correlates with auditory–motor synchronization, confers a learning benefit during PL. It is therefore likely that the dorsal prefrontal regions we report, which are shown to cohere in time with other frontoparietal regions, perform a role different from goal-oriented attention within the context of our tasks. This is in line with the idea that the same region can undertake different roles

depending on its interactions with other regions. It was not possible to determine the precise role of prefrontal regions alone from our data. On the other hand, we also show that the learning benefit relates to the way the frontoparietal network interacts with the auditory network. Another possibility, therefore, is that different kinds of dorsal prefrontal involvement during learning incur in either learning benefits or hindrance.

A possible reason for the lateralization discrepancies with Assaneo and colleagues [18] (bilateral engagement versus left lateralization) is the use of radically different measures and analyses (ICA of the BOLD signal versus a phase-locking value between an auditory stimulus and the MEG signals). Thus, although bilateral frontal and parietal regions may work together for synchrony (and learning benefits) in high synchronizers, as reflected in the ICA for fMRI analysis, each region may perform different computations to achieve that goal that are not captured by the PLV analysis, with entrainment in theta occurring only in left frontal regions. We similarly hypothesize that small structural differences (as those reported in [18], as captured by a particular method (diffusion-weighted MRI in that case) can give rise to large functional differences as appears to be the case in high synchronizers [32].

In sum, by considering individual differences in auditory–motor synchronization skills, our work sheds light onto the neural substrates of the SL of phonological word forms and shows that what appeared to be disparate results in the existing literature stems from pooling together fundamentally distinct populations. More specifically, we reveal that, beyond a universally recruited network for SWFL, an additional frontoparietal network that enables auditory–motor synchronization is selectively engaged by some individuals to produce a benefit in learning. The auditory–motor SSS test we use thus emerges, once more, as a useful tool to achieve a more nuanced characterization of speech related phenomena. This work, therefore, not only highlights the importance of considering individual differences in SL [41] but also sounds a note of caution about assuming the existence of monolithic mechanisms underlying such cognitive tasks.

## Methods

### Participants

A first cohort of 65 participants completed the behavioral protocol. Ten participants were removed because they spoke loudly instead of whispering or because they stopped whispering for longer than 4 sec (during the SSS test and/or the AS blocks). The data from 55 participants (34 females; mean age: 22; age range: 18 to 37) were analyzed.

A second cohort comprising 22 low and 22 high synchronizers participated in the fMRI protocol. Participants were selected from a larger group of 388 individuals, which completed the SSS test in the context of a previous study [18] (S1A Fig). Three participants were initially removed due to recording measurement error or problems completing the learning task (e.g., participant fell asleep, responses not recorded) and were removed from the fMRI sample. The final dataset thus comprised 41 individuals (23 females; mean age: 28; age range: 20 to 54; 19 high synchronizers and 22 low).

All participants were fluent English speakers with self-reported no neurological deficits and normal hearing. They provided written informed consent and were paid for taking part in the study. The local Institutional Review Board (New York University's Committee on Activities Involving Human Subjects) approved all protocols.

### Overall experimental design

The behavioral protocol consisted of 4 blocks of statistical word-form learning performed under 2 different conditions (PL and AS), followed by the SSS test (see Fig 5A). Four pseudo-

languages were generated, and their order was randomized across participants. In the SL blocks, participants were instructed to pay attention to the audio stream to be able to answer post-exposure questions about the perceived sounds. During the PL condition, participants passively listened to 2 of the pseudo-languages. During AS, participants repeatedly whispered the syllable "tah" while listening to the remaining 2 pseudo-languages. As in the SSS test, participants were *not* instructed to synchronize their speech to the auditory stimulus. Instead, they were told that the point of the whispering was to make the listening task more challenging. In line with the previous literature [9,20], we assumed that the effects of AS on SL would be due to an interference with the articulatory loop rather than to a higher executive load, which would be very unlikely given the highly automatized nature of the articulation subtask. PL and AS conditions were interleaved, and the 2 possible orders (PL–AS–PL–AS or AS–PL–AS–PL) were randomized across participants. After listening to each pseudo-language, learning was tested on a 2-alternative forced choice test.

The behavioral protocol was modified for fMRI acquisition. First, we divided the protocol into 2 experimental runs with one AS and one PL block each. In addition, 1 minute of rest was introduced before each statistical word learning block (PL or AS) and 2 minutes of speech production without auditory input (speech motor condition) were introduced at the end of each run (see Fig 5A). Specifically, the speech motor condition consisted in repeatedly whispering the syllable "tah" with no auditory input. The SSS test was not included in the fMRI session. Participants' speech synchronization abilities were assessed in a previous study [18].

Importantly, participants' whispered articulation was recorded during every AS block for both the behavioral and fMRI versions of the experiment (for the latter, we used an MRI compatible noise canceling microphone; OptoAcoustics FOMRI, Or Yehuda, Israel).

## Stimuli

For the SSS test and the word learning task, we created 4 pseudo-languages (L1 to L4) each containing 12 distinct sets of syllables (unique consonant-vowel combinations) handpicked to maximize both between and within set variability. The syllables in each pseudo-language were combined to form 4 distinct trisyllabic pseudo-words (henceforth, words). The words were relatively balanced on English word-average bi-phoneme and positional probabilities according to The Irvine Phonotactic Online Dictionary (IPhOD version 2.0; http://www.IPhOD.com), to maximize their learnability. Words were concatenated in pseudorandom order to form auditory speech streams with no gaps between words, lasting 2 minutes each. An equal number of nonconsecutive repetitions per word was ensured. For the learning test after each pseudo-language exposure, we created part-words by the concatenation of a word's final syllable and the first 2 syllables of another word of the same pseudo-language. A minute-long random syllable stream for the SSS test was created by the random combination of a set of 12 syllables different from those used for the pseudo-languages. The stream of syllables contained no pauses between them and no consecutive repetitions. Words, part-words and streams were converted to.wav files using the American Male Voice diphone database (US2) of the MBROLA text-to-speech synthesizer [50] at 16 kHz. All phonemes were equal in duration (111ms)—satisfying a constant syllable presentation rate of 4.5Hz, pitch (200Hz), and pitch rise and fall (with the maximum in the middle of the phoneme).

## Statistical word learning task

The statistical word learning task for each pseudo-language consisted of a learning phase, during which participants listened to the 2-minute-long streams containing the 4 words of the pseudo-language (L1 to L4); and a test phase, where each word of the pseudo-language was

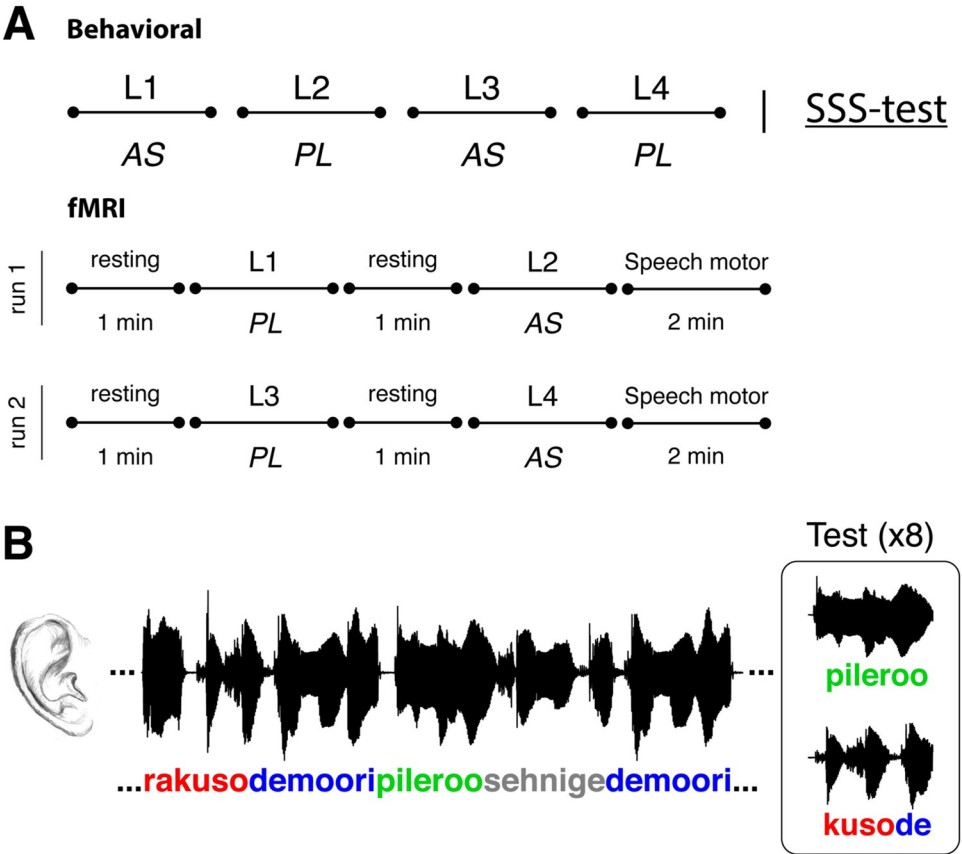

**Fig 5. Overall experimental design. (A)** The behavioral protocol consisted in 4 blocks of statistical word-form learning (each comprised a different pseudo-language) followed by the SSS test. The statistical word learning blocks were completed under 2 different conditions: PL, wherein participants passively listened to the pseudo-languages and AS, where participants concurrently, and repeatedly, whispered the syllable "tah". Conditions were interleaved and the order (AS–PL–AS–PL or PL–AS–PL–AS) was counterbalanced across participants. Lower panel: for the fMRI session, 2 speech motor and 4 rest blocks were added to the behavioral protocol. **(B)** Schematic representation of a statistical word learning block. Left panel: learning phase. Participants listened to the 2-minute-long auditory stream containing the 4 words of the pseudo-language. Right panel: test phase. Learning was assessed after each pseudo-language exposure by an 8 trial 2-alternative forced choice test contrasting a word, upper line, and a part-word, lower line. AS, articulatory suppression; PL, passive listening; SSS test, Spontaneous Speech Synchronization test.

presented against 4 part-words (randomly selected from the pool of 12 possible part-words) in a 2-alternative forced choice (see Fig 5B). During each test, words and selected part-words were presented twice each, within nonrepeating pairs, making this a total of 8 test trials. Test items were presented auditorily and in their written forms (left and right of the screen). Participants were required, for each test pair, to indicate their choice by pressing "1"/left or a "2"/ right according to the order of auditory presentation and location on the screen. The presentation of the pseudo-language*s* was counterbalanced between participants. In order to select the best phonology to orthography matching for the visual presentation, the written renderings of all words and part-words with the highest convergence among 5 independent native speakers were selected.

## SSS test

Participants in a sound isolated booth and seated in front of a computer with a microphone placed close to their mouth, listened to a rhythmic stream of syllables while whispering the

syllable "tah" concurrently and repeatedly for 1 minute. Next, they listened to isolated syllables and had to indicate for each one whether they were present in the stream. Participants were not explicitly instructed to synchronize their speech to the external audio; instead, the instruction was to correctly recall the syllables. Before performing this task, participants were primed to repeatedly whisper "tah" at a rate of 4.5 syllables/sec; the same rate at which the external syllables were presented. For further details about the test, see [18,51].

## Speech synchronization measurement

The degree of synchronization was measured by the PLV between the envelope of the produced speech and the cochlear envelope [23,52] of the rhythmic syllable stream or the pseudo-language. The PLV was computed using the following formula:

$$PLV = \frac{1}{T}|\sum_{t=1}^{T} e^{i(\theta_1(t) - \theta_2(t))}|,$$

where $t$ is the discretized time, $T$ is the total number of time points, and $\theta_1$ and $\theta_2$ the phase of the first and the second signals (i.e., of the envelopes of the auditory and produced speech signals). The PLV was computed for windows of 5 seconds length with an overlap of 2 seconds. The results for all time windows were averaged within each stimulus presentation, providing 1 PLV per block.

Envelopes were resampled at 100 Hz, filtered between 3.5 and 5.5 Hz, and their phases were extracted by means of the Hilbert transform.

## Definition of high and low synchronizers

Each participant was classified as a low or a high synchronizer according to their corresponding speech synchronization value (Whisper to Audio PLV) obtained during the SSS test. A threshold value was estimated from a previous dataset [18] comprising 388 PLVs obtained with different versions of the SSS test (S1A Fig). We applied a *k-means* clustering algorithm [53], using a squared Euclidean distance metric with 2 clusters, and computed the midpoint between the clusters' centers ($PLV_{threshold} = 0.49$). Participants with a PLV below/above this value were classified as low/high synchronizers.

## Scanning parameters

All fMRI scans took place at the NYU Center for Brain Imaging using a 3T scanner (Siemens Prisma 3T MRI scanner) and a 64-channel phased-array head coil. For the statistical word learning fMRI task, 2 runs of at least 400 (note that the test is self-paced and total timings vary between participants) sequential whole-brain multi-echo echo-planar imaging (EPI) volumes were acquired (TR = 1,500 ms, TE = 45 ms, flip angle = 77˚, voxel size = $2.0 \times 2.0 \times 2.0$ mm$^3$, 64 axial slices, acquisition size = 104×104). A high-resolution T1 MPRAGE image was also acquired (TR = 2,400 ms, TE = 2.24 ms, flip angle = 8˚, voxel size = $0.80 \times 0.80 \times 0.80$ mm$^3$, 256 sagittal slices, acquisition matrix = $320 \times 300$).

## fMRI and ICA preprocessing

Data were preprocessed using MATLAB R2018a and the Statistical Parameter Mapping software (SPM12, Wellcome Trust Centre for Neuroimaging, University College, London, United Kingdom, www.fil.ion.ucl.ac.uk/spm). For each participant, we first realigned the 2 word learning runs to the mean image of all EPIs. The T1 was then co-registered to this mean functional image and segmented using Unified Segmentation [54]. The deformation fields obtained

during the segmentation step were used to spatially normalize all functional images from each run to the Montreal Neurological Institute (MNI) template included in SPM12 (we maintained the original acquisition voxel size of $2.0 \times 2.0 \times 2.0$ mm$^3$). Images were finally spatially smoothed with a 6 mm FWHM kernel.

We used the GIFT [21] (v4.0b; http://mialab.mrn.org/software/gift) to apply group spatial ICA to the previously preprocessed fMRI data. Based on previous research in clinical and healthy populations [14,55,56], the number of independent components to be extracted was set to 20. Note that this is a purely data-driven approach. Data were intensity normalized, concatenated and, using principal component analysis, reduced to 20 temporal dimensions. Then, this preprocessed data were fed to the *infomax* algorithm [57]. The intensities of the spatial maps were in percentage of signal change after the intensity normalization, and thus no scaling was used.

To assess which of the 20 ICA networks retrieved were related to the different conditions of interest (PL and AS), both spatial and temporal classification methods were employed. First, for all participants, the spatial map of each individual ICA network was submitted to a second-level analysis using a 1-sample *t* test under SPM12 [58]. We then obtained, for each network, a group map of activity that was thresholded using a $p < 0.05$ family-wise error (FWE)-corrected threshold at the cluster level, with an auxiliary $p < 0.001$ threshold at the voxel level. Clusters with fewer than 50 voxels were not included in the analyses. We visually inspected these thresholded networks and 8 were discarded as they reflected artifacts related to movement or the presence of ventricles or blood vessels [14, 56].

Using GIFT, for the remaining 12 networks, we calculated a multiple regression that fitted each participant's network time-course to a model. The model was created using SPM12 by convolving the timing of both the main (AS and PL) and control (Rest and Speech motor) conditions with a canonical hemodynamic response. To control for motion artifacts, the model included 6 movement regressors obtained from the realignment step. The test phase was also included in the generalized linear model (GLM) model as a separate nuisance condition. Therefore, the beta values for the AS and PL conditions were computed from the part of the fMRI signal pertaining to the listening blocks (i.e., without the testing phase, which was modeled separately). By fitting a multiple regression between this model and each network's time-course, we obtained, for each condition, beta values that represented network engagement. For PL beta values, we used the rest condition as a baseline. For AS we used both the rest and the speech motor control conditions as baseline to capture the activity related to the learning process during AS itself and not to the motor activity related to the whispering. For any comparison using beta values, participants exceeding 2 SD were excluded from the analysis.

For the group spatial maps of each network, maxima and all coordinates are reported in MNI space. Anatomical and cytoarchitectonical areas were identified using the Automated Anatomical Labeling [59] and the Talairach Daemon [60] database atlases included in the xjView toolbox (http://www.alivelearn.net/xjview).

## Statistical analyses

Group analyses were performed on the proportion of correct responses averaged across same-type conditions (PL and AS). To test for differences between learning conditions and groups, we performed generalized linear mixed modeling in *R* (version 4.0.2) and *RStudio* (version 1.3.959) using the *lme4* package [61]. The dependent variable (responses to the learning tests) was assumed to have a binomial distribution and a logit link function was applied. An initial model included Condition (AS, PL), Group (High, Low) and their interaction as predictors. This model also included participants as a random effects factor to allow for varying intercepts

between participants. This was compared to alternative models with additional random effects factors Order, Language, and Trial number. These factors were removed, keeping the initial model, because the increase in variance explained by these more complex models was in all cases negligible. Akaike information criterion (AIC) was used for this assessment, thus selecting the model with the best balance between goodness of fit and complexity. The effects of the different predictors and their interactions on learning performance were assessed by means of likelihood ratio tests using the *afex* package [62] in *R*. These tests were based on Type 3 sums of squares. Following a significant interaction between Group and Condition, we estimated marginal means, using the *emmeans* package in R, of participants' performance within each group (Highs, Lows) for the PL and AS conditions. Where specified, we additionally used non-parametric Mann–Whitney–Wilcoxon and Wilcoxon signed-rank tests for between and within participant comparisons, respectively. Multiple comparisons were controlled using a FDR correction. Nonparametric Spearman rank correlations were used to assess the relationship between variables. Bayes factors ($BF_{01}$), which reflect how likely data are to arise from the null model (i.e., the probability of the data given H0 relative to H1), were also computed with the software JASP using default priors [63–65].

## Supporting information

**S1 Fig. Behavioral performance in the scanner.** **(A)** SSS test outcome. Histogram of 388 PLVs obtained in a previous work[1] with 2 different versions of the SSS test. Black dots represent the participants selected to complete the fMRI protocol. Black line represents the threshold value adopted in this work to separate high and low synchronizers: $PLV_{threshold}$ = 0.49. A *k-means* clustering algorithm using a squared Euclidean distance metric was applied over this distribution ($N$ = 388). The threshold value is the midpoint between the 2 clusters' centers. **(B)** Scatterplot displaying participants' PLV during AS inside the scanner as a function of the PLV from the SSS test. Red line represents the correlation of the data. The correlation is displayed for visualization purposes, to emphasize that the synchronization of low synchronizers is consistently worse than that of highs during the AS block. The correlation within groups remains significant only for high synchronizers ($r_{HIGH}$ = 0.45 $p_{HIGH}$ = 0.044; $r_{LOW}$ = 0.21 $p_{LOW}$ = 0.31). **(C)** Percentage of correct responses for the statistical word learning task during PL and AS conditions inside the scanner on the entire sample. **(D)** Percentage of correct responses for the statistical word learning task during PL and AS conditions inside the scanner for the low (blue color) and the high (orange color) synchronizers. The mixed-model analysis of this dataset yielded a significant difference between conditions (main effect of Condition (PL > AS), $\chi2$ = 5.40, $p < 0.05$)), and a main effect of group close to significance (Highs > Lows; $\chi2$ = 3.67, $p$ = 0.055), and a trending Condition*Group interaction ($\chi2$ = 2.74, $p$ = 0.098). Dots: model predicted group means. Bars: 95% confidence interval. Data for S1A and S1B Fig can be found in S5 Data. Data for S1C and S1D Fig can be found in S6 Data. AS, articulatory suppression; PL, passive listening; PLV, phase locking value; SSS test, Spontaneous Speech Synchronization test.
(DOCX)

**S2 Fig. Brain networks significantly activated during PL.** The different networks are shown over a canonical template with MNI coordinates on the upper portion of each slice. Neurological convention is used with a $p < 0.05$ FWE-corrected threshold at the cluster level and an auxiliary $p < 0.001$ threshold at the voxel level. In addition to the auditory (green) and frontoparietal (red) networks described in the main manuscript, a sensorimotor (magenta) and a right lateralized fronto-temporo-parietal (yellow) networks were also activated during PL. All these networks were significantly activated during PL for both high and low

synchronizers, except the frontoparietal, which was only active for the high. $^*p < 0.05$ Mann–Whitney–Wilcoxon between-group comparison, FDR corrected. Data for S2 Fig (plots) can be found in S7 Data. FDR, false discovery rate; FWE, family-wise error; MNI, Montreal Neurological Institute; PL, passive listening.
(DOCX)

**S3 Fig. Brain networks significantly activated during AS.** The different networks are shown over a canonical template with MNI coordinates on the upper portion of each slice. Neurological convention is used with a $p < 0.05$ FWE-corrected threshold at the cluster level and an auxiliary $p < 0.001$ threshold at the voxel level. In addition to the auditory (green) and frontoparietal (red) networks described in the main manuscript, a left (light blue) and a right (yellow) lateralized fronto-temporo-parietal network were also activated during AS. All networks were significantly activated during AS for both high and low synchronizers, except for the frontoparietal, which was only active for the high and the auditory, which was marginally significant for the lows ($p_{unc} = 0.03$). Data for S3 Fig (plots) can be found in S3 Data. AS, articulatory suppression; FWE, family-wise error; MNI, Montreal Neurological Institute.
(DOCX)

**S1 Data. Data for Fig 2A and 2B.**
(CSV)

**S2 Data. Data for Fig 2C and 2D.**
(CSV)

**S3 Data. Data for Fig 3A (bottom bar plot), Fig 3B, and S3 Fig (all plots).**
(CSV)

**S4 Data. Data for Fig 4B and 4C.**
(MAT)

**S5 Data. Data for S1A and S1B Fig.**
(CSV)

**S6 Data. Data for S1C and S1D Fig.**
(CSV)

**S7 Data. Data for Fig 3A (top bar plot), Fig 4A, and S2 Fig (all plots).**
(CSV)

## Author Contributions

**Conceptualization:** Joan Orpella, M. Florencia Assaneo, Pablo Ripollés, Diana López-Barroso, Ruth de Diego-Balaguer, David Poeppel.

**Data curation:** Joan Orpella, M. Florencia Assaneo, Pablo Ripollés, Laura Noejovich.

**Formal analysis:** Joan Orpella, M. Florencia Assaneo, Pablo Ripollés.

**Funding acquisition:** Ruth de Diego-Balaguer, David Poeppel.

**Investigation:** Joan Orpella, M. Florencia Assaneo, Pablo Ripollés.

**Methodology:** Joan Orpella, M. Florencia Assaneo, Pablo Ripollés.

**Project administration:** M. Florencia Assaneo, Ruth de Diego-Balaguer, David Poeppel.

**Resources:** Pablo Ripollés, David Poeppel.

**Software:** Joan Orpella, M. Florencia Assaneo, Pablo Ripollés.

**Supervision:** Ruth de Diego-Balaguer, David Poeppel.

**Validation:** Joan Orpella, M. Florencia Assaneo, Pablo Ripollés.

**Visualization:** Joan Orpella, M. Florencia Assaneo, Pablo Ripollés.

**Writing – original draft:** Joan Orpella, M. Florencia Assaneo, Pablo Ripollés, Diana López-Barroso, Ruth de Diego-Balaguer, David Poeppel.

**Writing – review & editing:** Joan Orpella, M. Florencia Assaneo, Pablo Ripollés, Diana López-Barroso, Ruth de Diego-Balaguer, David Poeppel.

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
