## [Editor Report · Decision Letter 0]

17 Dec 2021

Dear Dr Assaneo, 

Thank you for submitting your manuscript entitled "Population-level differences in the neural substrates supporting Statistical Learning" for consideration as a Research Article by PLOS Biology.

Your manuscript has now been evaluated by the PLOS Biology editorial staff, as well as by an academic editor with relevant expertise, and I am writing to let you know that we would like to send your submission out for external peer review.

Once your full submission is complete, your paper will undergo a series of checks in preparation for peer review. Once your manuscript has passed the checks it will be sent out for review. To provide the metadata for your submission, please Login to Editorial Manager (https://www.editorialmanager.com/pbiology) within two working days, i.e. by Dec 21 2021 11:59PM. 

**IMPORTANT: Please note that the editorial office will be closed from December 22nd to January 3rd. During this time, we will not be able to invite reviewers or take any action on your submission.

If your manuscript has been previously reviewed at another journal, PLOS Biology is willing to work with those reviews in order to avoid re-starting the process. Submission of the previous reviews is entirely optional and our ability to use them effectively will depend on the willingness of the previous journal to confirm the content of the reports and share the reviewer identities. Please note that we reserve the right to invite additional reviewers if we consider that additional/independent reviewers are needed, although we aim to avoid this as far as possible. In our experience, working with previous reviews does save time. 

If you would like to send previous reviewer reports to us, please email me at ggasque@plos.org to let me know, including the name of the previous journal and the manuscript ID the study was given, as well as attaching a point-by-point response to reviewers that details how you have or plan to address the reviewers' concerns. 

Given the disruptions resulting from the ongoing COVID-19 pandemic, please expect some delays in the editorial process. We apologise in advance for any inconvenience caused and will do our best to minimize impact as far as possible.

Kind regards,

Gabriel

Gabriel Gasque

Senior Editor

PLOS Biology

ggasque@plos.org

---

## [Decision Letter · Decision Letter 1]

14 Feb 2022

Dear Dr Assaneo,

Thank you for submitting your manuscript "Population-level differences in the neural substrates supporting Statistical Learning" for consideration as a Research Article at PLOS Biology. Your manuscript has been evaluated by the PLOS Biology editors, by an Academic Editor with relevant expertise, and by three independent reviewers. Please accept my apologies for the delay in sending the decision below to you.

In light of the reviews (below), we will not be able to accept the current version of the manuscript, but we would welcome re-submission of a much-revised version that takes into account the reviewers' comments. We cannot make any decision about publication until we have seen the revised manuscript and your response to the reviewers' comments. Your revised manuscript is also likely to be sent for further evaluation by the reviewers.

We expect to receive your revised manuscript within 3 months. 

**IMPORTANT - SUBMITTING YOUR REVISION**

*Re-submission Checklist*

*Published Peer Review*

*PLOS Data Policy*

*Blot and Gel Data Policy*

Sincerely,

Gabriel

Gabriel Gasque

Senior Editor

PLOS Biology

ggasque@plos.org

REVIEWS:

Reviewer #1: This study uses fMRI to investigate individual differences in language-related statistical learning. In a first step, participants were grouped into high and low synchronizers based on a Spontaneous Speech Synchronization test. In a subsequent word learning task, only the high synchronizers activated a fronto-parietal network and showed a behavioural benefit during passive listening compared to an active suppression task. 

My main question concerns the group difference between the low and the high synchronizers that also differ in the engagement of the fronto-parietal network. I wonder what the difference between the individuals in the two groups means: Is this a trait difference or simply a state difference, i.e., here most likely a different state of attention while listening? 

How does the fronto-parietal network identified in this study relate to previous work on involvement of the fronto-parietal network in attention and as a hub of cognitive control?

Could the authors provide more information about how participants were explicitly instructed in the different tasks? If I understand correctly, there was no task instruction about how to relate their whisper to the heard speech stream. About half of the participants aligned their whisper to the heard stream while the others did not. Would a plausible explanation of the observed results in this study be that participants in the "high" group, i.e. those that engage the fronto-parietal network more, perform a different task than those in the "low" group? 

Could the authors elaborate on their interpretation of the correlation between the engagement of the fronto-parietal network with the synchronization (PLV) between the perceived and produced syllables during AS in line 183/184: "suggesting a link between spontaneous auditory-motor synchrony and fronto-parietal network engagement". What does correlation across task performance mean? Is this correlation a marker for attentive participants?

In the second sample three of the behavioural effects were not significant, hence the formulation line 154 "this replication under notably adverse listening/learning conditions (i.e., during fMRI scanning) underscores the robustness of the behavioral results." should be adapted. 

Fig 4B: For Subject 2 the engagement of both networks is highly similar, whereas for S2 the networks are rather anti-correlated. The corresponding interpretation from the authors is "This suggests that the learning benefit shown by high synchronizers over lows was driven by a dynamic interplay in time between fronto-parietal and auditory networks." Could you elaborate more, on this and make clear why the strong alignment of both networks is interpreted as LESS dynamic interplay?

At the beginning it is stated "Yes - all data are fully available without restriction" and "All relevant data can be found in the paper's Supporting Information files." - However, there are no SI data files and at the end it is stated "DATA AND MATERIALS AVAILABILITY Correspondence and material requests should be addressed to M. Florencia Assaneo." This is inconsistent. 

Bar plot in Figures S2 and S3 are not specified - could individual data points be shown (e.g. violin plots) as in the paper it is repeatedly stated how important individual data are. 

Scatterplots in Figure 3b show the same data twice - why not color-coding the dots in the left panel for low and high synchronizers and remove the right panel?

Also in the scatterplots shown in Figure 4 it would be informative to colour-code the two groups of synchronizers.

In the scatterplot shown in Fig S1 B: this correlation seems to be driven by the group differences (could you report the within group correlations in addition).

Reviewer #2: The authors have previously shown that statistical language learning correlates with participant's auditory-motor synchronization skills, which can be measured with a simple spontaneous speech synchronization (SSS) task. In the new study they replicate and extend this highly interesting finding by using fMRI to investigate the neural bases of individual differences in SSS and statistical learning. The results show that high synchronizers perform better in statistical language learning (replication) than low synchronizers and that they also activate additional fronto-parietal neural network when listening to syllable streams. All participants activated an auditory network. Overall, this is an elegant study that provides new insight into the neural basis of statistical language learning and individual differences, which has been poorly understood. Statistical learning is a fundamental cognitive capacity that is under active investigation among cognitive scientists. I therefore believe that these new findings will be interesting to a wide audience.

1. Specificity vs. domain-generality. Statistical learning is a domain general learning mechanism, although it is often investigated using speech stimuli. In my opinion the main weakness of the current study is the lack of control conditions/non-speech stimulus streams. The authors link they results tightly to language learning and phonology, although there is there is no direct evidence that the enhanced performance in the SSS task and/or activation of the fronto-parietal network is specific to speech/language. The lack of control stimuli limits the interpretation of the findings and the authors should acknowledge these limitations. 

2. Laterality. In their previous paper in Nature Neuroscience the authors reported left-lateralized white-matter and neural entrainment (MEG) differences between high and low synchronizers. In the new study the additional fronto-temporal network seems to be bilateral. This is an intriguing difference between the studies, which would deserve proper discussion.

Reviewer #3: In this manuscript, Orpella and colleagues elegantly reconcile apparently disparate neuroimaging findings within the auditory-linguistic statistical learning literature. They build off of prior work demonstrating a link between word segmentation abilities and spontaneous speech auditory-motor synchronization, suggesting that only a subset of learners (high auditory-motor synchronizers) recruit a fronto-parietal network during learning. These same high synchronizers are also most affected when performing a concurrent articulatory suppression task. This manuscript is really strong work—thoroughly motivated, well-written, and the figures were a standout. However, I do feel that its impact could be highlighted to a greater extent, particularly given that a link between high synchronizers and statistical word learning has already been observed. I fully agree that "this work sounds a note of caution about assuming the existence of monolithic mechanisms underlying cognitive tasks" (287-289). Along these lines, the manuscript would be enhanced by more specific, substantial descriptions of how these findings add to the existing literature on individual differences in SL.

MAJOR

1. 274-277: "Broadly speaking, therefore, our results line up with recent theories of SL, which postulate the works of both learning systems (e.g., comprising auditory areas in the case of auditory input) and modulatory attentional/control systems (e.g., as supported by fronto-parietal networks) underlying learning performance." 

Given the observed involvement of prefrontal cortex, I recommend richer discussion of how this work relates to proposals that children and adults might recruit different neural learning systems and that it may be worth considering downsides to a mature prefrontal cortex. These very recent refs might be useful:

Smalle et al. (2021). Less is more: Depleting cognitive resources enhances language learning abilities in adults. Journal of Experimental Psychology: General, 150(12), 2423-2434. 

Smalle et al. (2022). Unlocking adults' implicit statistical learning by cognitive depletion. PNAS, 119 (2), e2026011119.

2. Fig. 4; 186-210: The incorporation of behavioral data into the neuro analysis is a strength of this paper. However, it is worth noting that there is a substantial push in the SL literature towards more effectively capturing individual differences in SL.

Siegelman, N., Bogaerts, L., & Frost, R. (2017). Measuring individual differences in statistical learning: Current pitfalls and possible solutions. Behavior Research Methods, 49(2), 418-432.

The present work involves tests of SL with few trials, test items of equivalent difficulty, and, in the case of the learning benefit analysis, difference scores (see discussion in Trafimow (2015)—A defense against the alleged unreliability of difference scores). Can the authors discuss/ demonstrate the reliability of the behavioral measures used?

MINOR

3. 313-315: "It is assumed that the executive load imposed by the repeated syllable articulation in this paradigm is minimal, given the highly automatized nature of the articulation subtask." Can the authors provide a stronger justification for this?

4. 410-450: It's not clear me whether the test phase was included when convolving the timing of the AS and PL conditions.

---

## [Decision Letter · Decision Letter 2]

27 May 2022

Dear Dr Assaneo,

Thank you for your patience while we considered your revised manuscript "Population-level differences in the neural substrates supporting Statistical Learning" for publication as a Research Article at PLOS Biology. This revised version of your manuscript has been evaluated by the PLOS Biology editors, the Academic Editor, and one of the original reviewers.

Based on the the reviewer feedback and our Academic Editor's assessment of your revision, we would like to move towards acceptance of this manuscript for publication. At this stage, we simply need you to satisfactorily address a few editorial issues, and some data and other policy-related requests (listed at the bottom of this email).

To ensure your work can be fully appreciated by our broad readership, we request that you consider a change to the title and some modifications to your abstract.

**Suggested new title:

Differential activation of a front-parietal network explains population-level differences in language learning.

**Modified abstract:

People of all ages display the ability to detect and learn from patterns in seemingly random stimuli. Referred to as statistical learning, this process is particularly critical when learning a spoken language, helping in the identification of discrete words within a spoken phrase. Here, by considering individual differences in speech auditory motor synchronization, we demonstrate that recruitment of a specific neural network supports behavioral differences in new language acquisition. While independent component analysis of fMRI data revealed that a network of auditory and superior pre/motor regions is universally activated in the process of learning, a fronto-parietal network is additionally and selectively engaged by only some individuals. Importantly, activation of this fronto-parietal network as related to a boost in learning performance, and interference with this network via articulatory suppression (i.e. producing irrelevant speech during learning) normalizes performance across the entire group. Our work provides novel insights on language-related statistical learning and reconciles previous contrasting findings. These findings also highlight a more general need to factor in fundamental individual differences for precise characterization of cognitive phenomena. 

We expect to receive your revised manuscript within two weeks. 

*Published Peer Review History*

*Press*

Sincerely,

Kris

Kris Dickson, Ph.D. (she/her),

Neurosciences Senior Editor/Section Manager,

kdickson@plos.org,

PLOS Biology

DATA POLICY:

Fig 2A-D ; Fig 3Agraphs & B ; Fig 4A-C graphs

Supplemental Fig 1A-D; SFig 2graphs; SFig 3 graphs

Please also ensure that figure legends in your manuscript include information on where the underlying data can be found (THIS STEP IS OFTEN FORGOTTEN!), , and ensure your supplemental data file/s has a legend.

CODE POLICY:

https://journals.plos.org/plosbiology/s/materials-software-and-code-sharing

We expect that all researchers submitting to PLOS submissions in which software is the central part of the manuscript will make all relevant software available without restrictions upon publication of the work. Authors must ensure that software remains usable over time regardless of versions or upgrades. If the original software is not able to be shared, authors must provide a reasonable facsimile.

DATA NOT SHOWN?

Reviewer remarks:

Reviewer's Responses to Questions

PLOS authors have the option to publish the peer review history of their article (what does this mean?). If published, this will include your full peer review and any attached files.

Reviewer #1: No

---

## [Editor Report · Decision Letter 3]

14 Jun 2022

Dear Dr Assaneo,

Thank you for the submission of your revised Research Article "Differential activation of a fronto-parietal network explains population-level differences in statistical learning from speech" for publication in PLOS Biology. On behalf of my colleagues and the Academic Editor, Matthew Rushworth, I am pleased to say that we can in principle accept your manuscript for publication, provided you address any remaining formatting and reporting issues. These will be detailed in an email you should receive within 2-3 business days from our colleagues in the journal operations team; no action is required from you until then. Please note that we will not be able to formally accept your manuscript and schedule it for publication until you have completed any requested changes.

PRESS

Sincerely, 

Kris

Kris Dickson, Ph.D. (she/her)

Neurosciences Senior Editor/Section Manager

PLOS Biology

kdickson@plos.org